# Interpreting deep embeddings for disease progression clustering

**Anna Munoz-Farre** [* 1] **Antonios Poulakakis-Daktylidis** [* 1] **Dilini Mahesha Kothalawala** [1]
**Andrea Rodriguez-Martinez** [1]

## Abstract

We propose a novel approach for interpreting deep embeddings in the context of patient clustering. We evaluate our approach on a dataset of participants with type 2 diabetes from the UK Biobank, and demonstrate clinically meaningful insights into disease progression patterns.

## 1. Introduction

The advent of transformer-based models has revolutionised the field of natural language processing (Vaswani et al., 2017). These models have shown significant potential in healthcare applications, where large volumes of structured data (disease diagnoses, medication prescriptions, surgical procedures, laboratory results, etc) are collected and stored in the form of electronic health records (EHR) (Li et al., 2020; Steinberg et al., 2021; Yang et al., 2022). This has enabled researchers to extract insights into the underlying mechanisms that drive disease progression, as well as to cluster patients based on their particular disease profile and comorbidities (Hassaine et al., 2020; Landi et al., 2020; Lee & van der Schaar, 2020; Rasmy et al., 2021). In recent years, there has been an increase in prioritising and establishing better benchmarks and developing more reliable and trustworthy models (Meng et al., 2022). The interpretability of such models is crucial to identify potential biases and ensure fairness when applying such models in the healthcare context, which will also have to go through regulatory approvals before using them on real patients (Kumar et al., 2022; Ghaffar Nia et al., 2023; Lahav et al., 2019).

In this paper, we propose a new method for disease progression clustering, using transformer-based embeddings derived from large-scale structured EHR data. We define a framework to clinically interpret the learnt embeddings, which enables us to identify disease progression stages. Finally, we apply time-series clustering to stratify patients into clinically-relevant subgroups with different aetiological and prognostic profiles. We validate our approach by showing that the embedding space is associated with disease-specific clinical themes, with patients progressing across them.

The contributions of our paper are:

- **(i)** a method for interpreting the embedding space in the clinical setting (Section 3.3)

- **(ii)** the presentation of a patient clustering method based on disease trajectories learned from embeddings (Section 3.4)

- **(iii)** an in-depth clinical evaluation for each disease stage and cluster (Sections 4.3 and 4.4).

## 2. Related Work

### 2.1. Intepreting deep embeddings

Interpreting deep embeddings in language models has been a subject of extensive research. Visualization techniques, such as t-SNE (van der Maaten & Hinton, 2008) or UMAP (McInnes et al., 2018), have been used to reveal semantic relationships and analogies between words (Lal et al., 2021). Most popular methods focus on learning about feature importance and feature interaction for each prediction (Lundberg & Lee, 2017; Chen et al., 2018; Crabbé et al., 2020; Tsang et al., 2018; Shrikumar et al., 2019; Ribeiro et al., 2016).

In the clinical setting, Schulz et al. (2020) have proposed an explanation space constructed from feature contributions for inferring disease subtypes. Bai et al. (2018) were among the first to account for different temporal progression of medical conditions and add an interpretability aspect on top of RNNs. Med-BERT touched on the interpretability aspect as well by visualising attention patterns of the model (Rasmy et al., 2021). Rao et al. (2021) used a transformer-based architecture coupled with perturbation techniques to identify clinically explainable risk factors for heart failure.

*Equal contribution [1]BenevolentAI, London, UK. Correspondence to: Anna Munoz-Farre <anna.munoz.farre@benevolent.ai>, Antonios Poulakakis-Daktylidis .

*Workshop on Interpretable ML in Healthcare at International Conference on Machine Learning (ICML)*, Honolulu, Hawaii, USA. 2023. Copyright 2023 by the author(s).

## 2.2. Disease progression clustering

Giannoula et al. (2018) applied dynamic time warping directly on time sequences of ordered disease codes. Zhang et al. (2019) used the hidden state of LSTM layers as time-series for subtype identification, and dynamic time warping for computing similarities, focusing on Parkinson's disease. More recently, Lee & van der Schaar (2020) applied temporal clustering based on future outcomes using an actor-critic architecture with an RNN as an encoder, and multiple loss functions to induce embedding separation and cluster purity.

## 3. Methods

### 3.1. Defining clinical histories through EHR

Medical ontologies are the basic building block of how structured EHR data are recorded. They are hierarchical data structures which contain healthcare concepts that enable healthcare professionals to consistently record information. Ontology concepts are composed of a unique identifier and a corresponding human-friendly description (for example, *J45-Asthma* is a code-description pair in the ICD10 ontology used in hospitalization EHR). However, each healthcare setting (e.g. primary care, secondary care) uses a different ontology (NHS), which means a single patient might have their records in multiple ontologies.

For each patient, we defined their entire clinical history as the concatenation of sequences of ontology text descriptions $(\xi_{\theta_1}, \ldots, \xi_{\theta_t})$, $\xi_{\theta_i} \in \Xi_\Theta$, $i = 1, \ldots, t$, ordered over time (Munoz-Farre et al., 2022) across all EHR sources, with $\Xi_\Theta$ being the set of descriptions for each ontology $\theta$. To capture temporal patterns and changes in disease progression, we sliced each patient's history into "snapshots" around the date of diagnosis (Figure 1). Snapshot length was chosen based on the available dataset and the disease use-case. For each snapshot, we processed the raw sequence of textual descriptions into tokens (word and sub-word pieces), using a tokenizer $W$ as $X = W(\xi_{\theta_1}, \ldots, \xi_{\theta_t}) = (x_1, \ldots, x_n)$, with $n$ as the tokenized sequence length.

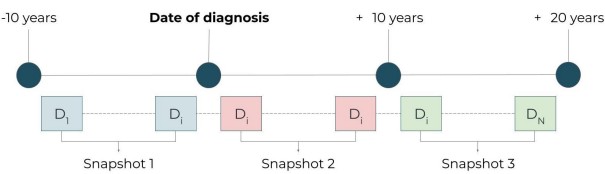

*Figure 1.* Example of constructing snapshots from EHR data with a ten year window.

### 3.2. Model design

We trained a model that classifies disease status based on EHR sequences. Let $X^{(p,s)} = (x_1^{(p,s)}, \ldots, x_n^{(p,s)})$ denote the tokenized input sequence of an individual $p$ and a snapshot $s$. It forms the input to an encoding function $\mathbf{e}_1^{(p,s)}, \ldots, \mathbf{e}_n^{(p,s)} = Encoder(X^{(p,s)})$, where each $\mathbf{e}_i$ is a fixed-length vector representation of each input token $x_i$. Let $\mathbf{y}^{(p)} \in \{0,1\}$ be the disease label. To calculate disease probability $\mathbf{P}(\mathbf{y}^{(p,s)}|X^{(p,s)})$, the embeddings of the CLS token are fed into a decoder $z_1^{(p,s)}, \ldots, z_D^{(p,s)} = Decoder(\mathbf{e}_1^{(p,s)}, \ldots, \mathbf{e}_n^{(p,s)})$, and the resulting logits are fed into a softmax function $\sigma$ $\mathbf{P}(y^{(p,s)}|\mathbf{e}_1^{(p,s)}, \ldots, \mathbf{e}_n^{(p,s)}) = \sigma(z^{(p,s)})$ (Figure 2).

### 3.3. Embedding space interpretation framework

Patient snapshots fed into the model represent different disease stages, so we expected the resulting embedding space to reflect them. To demonstrate this, we reduced the normalized embeddings generated by the transformer-based encoder for each sequence to two-dimensional vectors $U^{(p,s)} = (u_1^{(p,s)}, u_2^{(p,s)})$, using the Uniform Manifold Approximation and Projection (UMAP) algorithm (McInnes et al., 2018) (Figure 2).

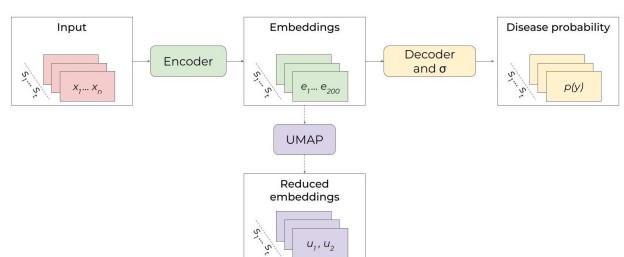

*Figure 2.* Model diagram flow. Snapshot sequences are tokenized to generate the input, which is fed into the encoder. The embeddings of the CLS token are then fed into a linear decoder and through a softmax function to get disease probability. After the model is trained, the embeddings are reduced to two-dimensional vectors, using UMAP.

To evaluate separation of disease stages in the embedding space, we examined the correlation between the reduced embeddings $U$ and other available clinical markers $F = (f_1, \ldots, f_k)$. We included clinically-relevant markers extracted from snapshots of EHR data, such as laboratory tests, medication prescription, other co-occurring conditions (comorbidities), etc. Specifically, we computed the point-biserial correlation coefficient (Lev, 1949) between each patient's reduced embeddings $U^{(p,s)}$ and their comorbidities, and medication prescription. We calculated the L2 norm (Euclidean distance to the origin) for each clinical

marker $f_k$ as $d_{f_k} = \sqrt{r^2_{f_k,u_2} + r^2_{f_k,u_1}}$, 0 being no correlation between $f_k$ and $(u_1, u_2)$. We then evaluated whether the most highly-correlated conditions and medications were specific to the disease in question, and whether we could identify different clinical themes (Figure 3).

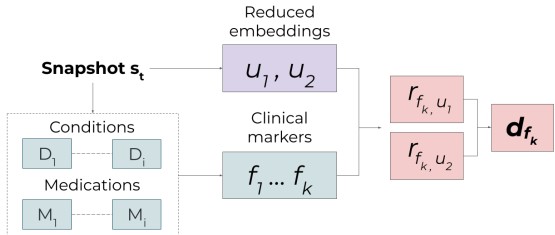

Figure 3. Workflow to find the correlation between the reduced embeddings and clinical markers for each snapshot $s_t$, using the Point-biserial correlation coefficient $r_{f_k,u_i}$, and calculating the L2 norm (distance to 0), $d_{f_k}$.

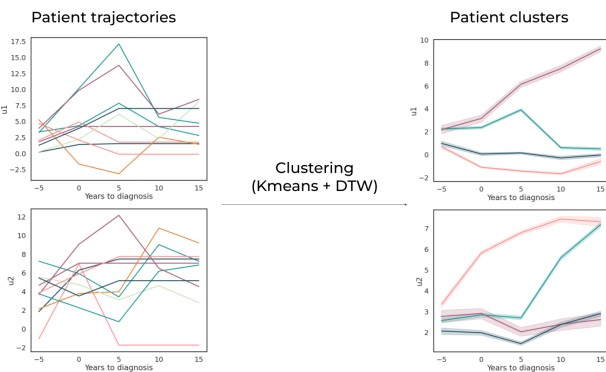

Figure 4. Diagram of patient clustering on trajectories (with an example 5 year time step). We first reduce the embeddings for each snapshot using UMAP (left). We then perform time-series clustering using the k-means algorithm with multivariate dynamic time warping (DTW) on the reduced trajectories (right). Note that patients shown in the left figure are simulated for data protection purposes.

## 3.4. Patient clustering

For each patient, we have multiple snapshots in the form of reduced two-dimensional embeddings $(u_1, u_2)$, which can be used as time series data to study patient trajectories in the embedding space. We aligned patients using linear interpolation, excluding those with less than three snapshots. We performed temporal clustering of patients using the k-means algorithm with multivariate dynamic time warping (DTW) (Müller, 2007) (Figure 4). Finally we used the embedding interpretation framework proposed in the previous section to clinically characterize each patient cluster.

## 4. Experiments and Results

### 4.1. Defining study population: Type 2 Diabetes cohort

This research has been conducted using the UK Biobank (UKBB) Resource under Application Number 43138, a large-scale research study of around 500k individuals (Sudlow et al., 2015). It includes rich genotyping and phenotyping data, both taken at recruitment and during primary care (general practice, GP) and secondary care (admitted hospitalizations) visits. To avoid bias or stratification based on data source, we restricted the dataset to individuals that have both primary and secondary care data linked, which are coded using the read and ICD ontologies, respectively (NHS). The final cohort includes 154, 668 individuals.

Type 2 diabetes mellitus (T2D) is one of the most prevalent chronic diseases worldwide, and patients are primarily diagnosed and managed in primary care. It presented an excellent use-case for our framework, because we have orthogonal data available to evaluate the embedding space (such as

medication prescription and other co-occurring conditions). Our attempt was to identify known T2D epidemiological associations and clinical markers in the patients with T2D (Zghebi et al., 2020; Pearson-Stuttard et al., 2022).

We selected a cohort of 20.5k patients with T2D (cases) and a corresponding cohort of 20.5k control patients (matched on biological sex and age). We cleaned inconsistent diabetes mellitus codes from cases, and removed type 1 diabetes patients from controls A.1.2. Both ICD and Read ontologies are structured in a hierarchy, so we took the parent T2D code-descriptions for hospital and GP, and all of their children. We removed them from all input sequences, to force the model to learn disease relevant history representations without seeing the actual diagnosis. We spliced each patient's history into three time snapshots of 10 years around diagnosis: [-10,0,10,20], where 0 is date of diagnosis (more details in Appendix A.1, and Figure 8).

### 4.2. Model training

Using the full UKBB dataset, we first trained a BertWordPieceTokenizer, resulting in a vocabulary size of 2025 tokens. We then trained a transformer-based encoder with a hidden dimension of 200 on the Masked Language Modeling (MLM) task (Devlin et al., 2019), to learn the semantics of diagnoses. The proposed classifier uses the trained encoder and a fully connected linear layer as the decoder. To be able to use the embeddings of all T2D patients, we trained a total of five models in a cross-validation fashion (more details in Appendix A.2). All results presented are predictions and embeddings of each model on its respective independent test set. We evaluated model performance on

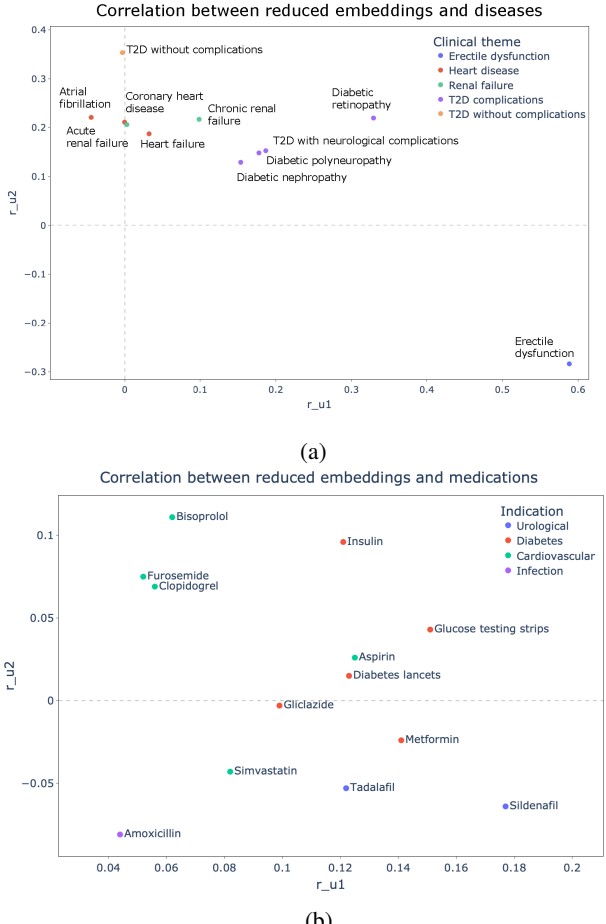

(a)

(b)

*Figure 5.* Associations between two-dimensional reduced embeddings and clinical factors. 5a Association with diseases, where colours indicate broad disease theme. 5b Association with medication, where colours indicate broad indication disease theme.

the test set of each fold using standard metrics for binary classification, with an average recall of 0.92 and precision of 0.82 across sequences.

### 4.3. Embedding space interpretation

We used the default UMAP hyperparameters to reduce the embeddings to two-dimensional vectors, after experimenting with different combinations (Appendix A.3.1). We then examined the most strongly-correlated clinical markers by extracting the highest-ranked comorbid-diseases (Table 1, Figure 5a) and medications (Table 2, Figure 5b). To show unique clinical markers, we mapped all conditions (from GP or hospital) to the ICD10 ontology and medications to the Anatomical Therapeutic Chemical (ATC) Classification System (WHO).

We evaluated clinical themes, in terms of T2D management, comorbidities and complications:

- **T2D management**: Metformin is associated with $u_1$, which is the preferred initial glucose-lowering medication for most people with T2D. We also find gliclazide, which can be used instead of or in combination with metformin, and diabetes lancets and glucose testing strips, which are used to test blood glucose levels. Interestingly, insulin is strongly associated with $u_2$, which is given to severe T2D patients (Medscape, b).

- **Comorbidities**. We find two main clinical themes. **Cardiovascular disease** (CVD) is associated with $u_2$. T2D patients have a considerably higher risk of cardiovascular morbidity and mortality, due to high blood sugar levels causing blood vessel damage and increasing the risk of atherosclerosis (Einarson et al., 2018). Moreover, hypercholesterolemia and high LDL cholesterol, which are strongly associated with T2D, are risk factors for CVD. When looking at medication, we find furosemide and bisoprolol, which are used to manage heart failure (HF) (Medscape, d), and antiplatelet agents, such as clopidogrel or aspirin, given to patients with coronary heart disease (CHD) (Medscape, a).

  **Erectile dysfunction** (ED) is a prevalent comorbidity in male T2D patients (MacDonald & Burnett, 2021), and is managed with drugs such as tadalafil (Cialis) and sildenafil (Viagra) (Medscape, c), which are all associated with $u_1$.

- **T2D complications**: Even though all T2D related ontology terms were excluded from the input data, the model learned to separate T2D patients without complications to those with complications, which are associated with both $u_1$ and $u_2$, such as diabetic retinopathy, nephropathy, or polyneuropathy (Cheung et al., 2010). Moreover, T2D is a leading risk factor for chronic kidney disease and renal failure, which is found in the same area (McGill et al., 2022).

### 4.4. Patient clusters evaluation

We used linear interpolation with a five year step to align patients' snapshots, resulting in the following time points relative to the date of diagnosis: [-5,0,5,10,15]. We found four patient clusters (experiments in Appendix A.3.3, with demographics and age of diagnosis in Table 3). When examining patient progression across the embedding space (Figure 6), we observed that patients start in the same space (with no diagnosis of T2D), and move towards clinical themes, corresponding to what we saw in Figure 5a.

To look at comorbidity progression, we calculated prevalence of the most strongly correlated themes, looking at how many patients had at least one diagnosis of the theme for each group and time point (Figure 7). Starting from the lowest $u1, u2$, we see that patients in cluster 3 remain

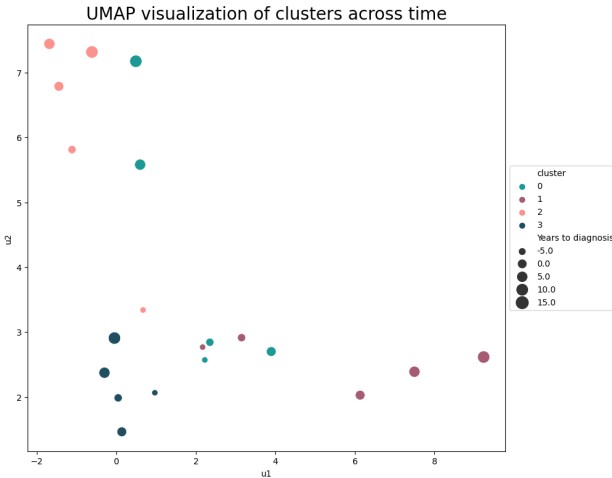

*Figure 6.* UMAP visualization of 4 clusters (mean per cluster and time window). Colour indicates different clusters, and size indicates time windows (the smallest is 5 years before diagnosis, and the largest is 15 years after diagnosis.)

in the initial area, indicating they might be in a controlled disease state. Cluster 2 is a slightly older population, that moves towards the cardiovascular and T2D without complications area. Following closely, cluster 0 represents a more severe group, with a combination of high prevalence of cardiovascular disease, renal failure and T2D complications. Finally, cluster 1 represents mostly male patients with T2D complications and erectile dysfunction.

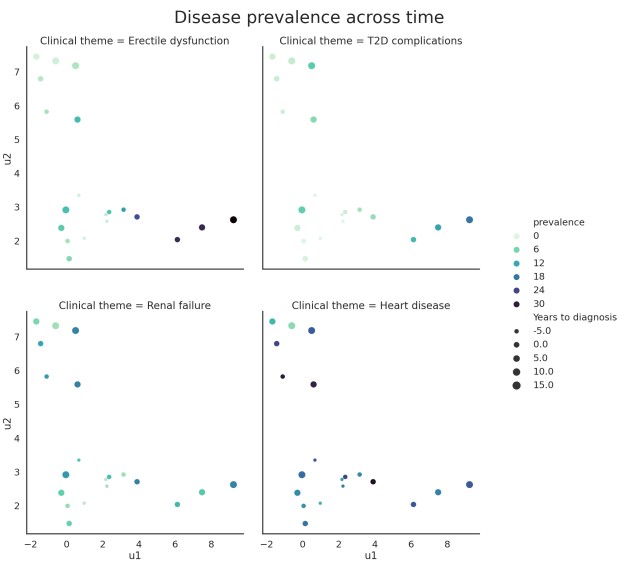

*Figure 7.* Disease theme prevalence for each cluster and snapshot. Prevalence increases over time (darker colour) for in each cluster.

## 5. Conclusions

Here, we proposed a framework for interpreting the embedding space of transformer-based models in a clinically-meaningful way. We showed that the model learnt to distinguish disease-specific clinical themes and we validated that by replicating associations with known T2D comorbidities, complications, and medications. We performed temporal clustering of patients and identified distinct and clinically interpretable disease progression patterns. Our framework can be adapted to any disease use-case, and any available clinical dataset. It can be used to identify disease-specific, clinically and biologically relevant groups to personalize treatment and interventions for patients.

## 6. Acknowledgements

This research has been conducted using the UK Biobank Resource under Application Number 43138. Using real patient data is crucial for clinical research and to find the right treatment for the right patient. We would like to thank all participants who are part of the UK Biobank, who volunteered to give their primary and secondary care and genotyping data for the purpose of research. UK Biobank is generously supported by its founding funders the Wellcome Trust and UK Medical Research Council, as well as the British Heart Foundation, Cancer Research UK, Department of Health, Northwest Regional Development Agency and Scottish Government.

We are particularly grateful to Aylin Cakiroglu, Prof. Spiros Denaxas, Rogier Hintzen, Nicola Richmond, Ana Solaguren Beascoa-Negre, Benjamin Tenmann, Andre Vauvelle, and Millie Zhao for their feedback, insightful comments and the many inspiring conversations.

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

# A. Appendix.

## A.1. Data processing

### A.1.1. UKBB

The UK Biobank (UKBB) (Sudlow et al., 2015) is a large-scale research study of around 500k individuals between the ages of 40 and 54 at the time of recruitment. It includes rich genotyping and phenotyping data, both taken at recruitment and during primary and secondary care visits (GP and hospital). We used patient records visits in the form of code ontologies Read version2/ Clinical Terms Version 3 (GP), and ICD-9/10 (hospital) together with their textual descriptions. We restricted the data set to individuals that had both hospital and GP records, reducing the cohort to 154, 668 individuals. Requiring individuals to have entries in their GP records reduces bias towards acute events that usually present in hospitals, but we note that removing individuals without any hospital records may still bias the data towards more severe cases.

A patient can be admitted to the hospital for multiple days. We treated an entire hospital admission as one point in time using the admission date, and only kept unique ICD-10/ICD-9 codes for each visit. We aggregated visits that were less than a week apart into one visit keeping only unique codes. This approach removed repeated codes, thus avoiding redundancy and reducing sequence length.

### A.1.2. TYPE 2 DIABETES COHORT EXTRACTION

When extracting the T2D cohort, we noticed that some patients had both a diagnosis for type 1 and type 2 diabetes, or had an undefined diabetes mellitus diagnosis. This mistake might happen, for example, when admitting patients in the hospital without looking at their entire clinical history. To properly label those patients, we looked at the medications that those patients were taking, identifying the ones that are given to type 2 diabetes patients (Medscape, b). In those cases where it was unclear, we dropped the patients from the cohort. Finally, we did not include type 1 diabetes patients in the control cohort. Our final cohort included 20.5k patients with T2D and 20.5k control patients (matched on biological sex and age).

T2D is a chronic progressive condition, so that taking 10 years for each snapshot was enough to represent disease stage without losing important information. There were some patients that had longer snapshots than the maximum sequence length (64) after tokenization, so we split those sequences. The result was that a patient could have multiple snapshots for each given 10 year window (Figure 8). For each input sequence, we calculated a mean time to T2D diagnosis by examining the first and last date present in that sequence. This way we had each snapshot associated to a given time point (relative to date of diagnosis), which could then be used for interpolation and patient clustering, as explained in Section 3.4.

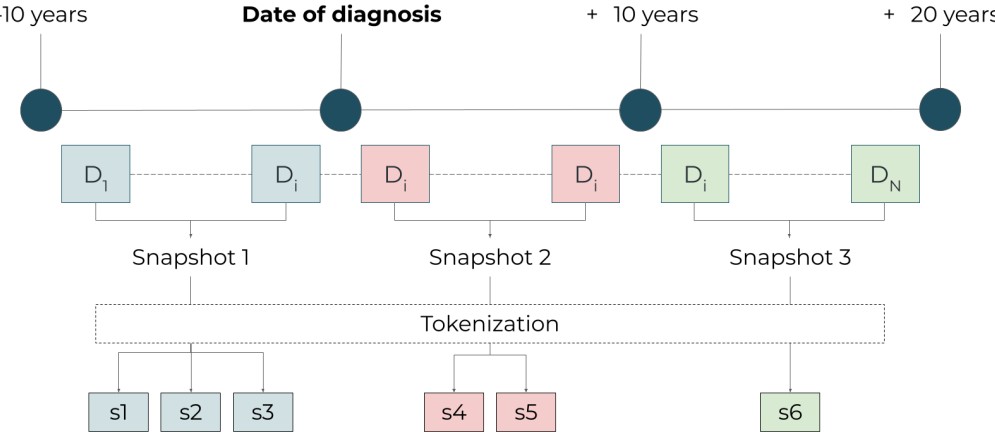

*Figure 8.* Example of cutting a patient's history into 10-year snapshots. Those sequences that were longer than the maximum sequence length after tokenization were split into multiple sequences, resulting in the final used snapshots.

## A.2. Model training

To train on the classification task, we split our data set into five equally sampled folds $f_0,...f_4$, containing unique patients. To be able to use the embeddings of all T2D patients, we trained a total of five classification models on three folds, holding back folds $f_i$ for validation and $f_{(i+1) \mod 5}$ for testing for model $i$, $i = 1, \ldots, 5$. This maintained a $60/20/20$ training, validation and testing split. Each model was trained for 30 epochs, batch size of 64, learning rate of $10^{-5}$, and a warm-up proportion of $0.25$, using gradient descent with AdamW optimizer, weight decay of $0.01$ and early stopping. Performance was monitored every $0.25$ epochs on the validation fold for both recall and precision.

## A.3. Dimensionality reduction and patient clustering

### A.3.1. UMAP HYPERPARAMETERS

We used UMAP to reduce our embeddings to a two-dimensional vector. There are several hyperparameters that can be tuned that might affect the results and final patient clusters. We experimented with the number of nearest neighbours (*n_neighbors* =[15, 30, 50, 100]) and minimum distance (*min_dist*=[0.01, 0.1, 0.5, 1]) (McInnes et al., 2018). For the same number of clusters $k$, we wanted to verify that the resulting patient clusters were maintained (robust to different hyperparameters). As a metric, we used euclidean distance on the normalized embeddings. Our hypothesis was that clusters are maintained, so we looked at the overlap of patient groups across combinations, and calculated the Jaccard similarity score. We saw that there are 3 very clean clusters that are always found, and two others with a higher overlap, but highly separated from the first 3 (Figures 9a and 9b). After seeing that clusters were robust across combinations, we decided to use the recommended UMAP hyperparameters in the original implementation (*n_neighbors* = 15 and *min_dist* = 0.1).

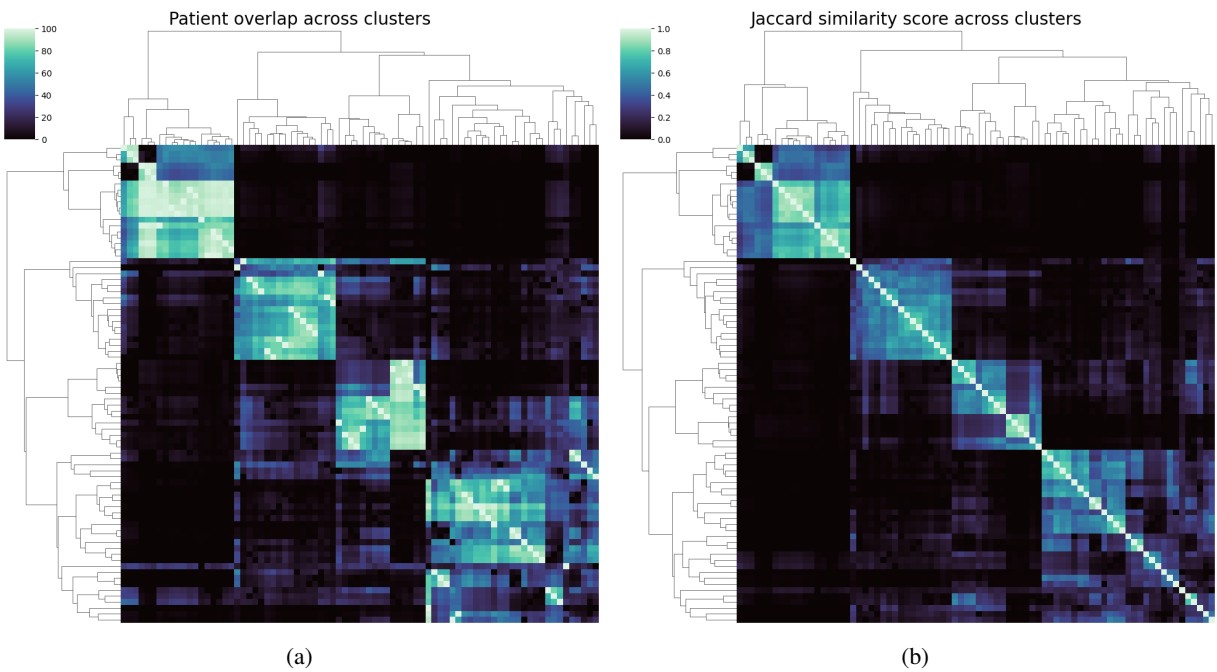

(a)  (b)

*Figure 9.* Clustermap of metrics to compare patient clusters across different UMAP hyperparameters combinations. 9a Clustermap of the patient overlap. 9b Clustermap of the Jaccard similarity score.

A.3.2. Embedding space interpretation

For each patient snapshot, we looked at all diagnoses in primary and secondary care, and medication prescriptions given in primary care in their clinical history. Note that type 2 diabetes associated diagnoses (including complications) were excluded from the input data, but we included them in these analysis to see whether the model learnt certain characteristics from the rest of the sequence. We looked at the most strongly correlated clinical markers by calculating the L2 norm (euclidean distance to origin), and took the top 15. We dropped duplicated diagnoses that were essentially the same condition, keeping the most correlated one (for example, *Atrial fibrillation and flutter* was dropped and *Atrial fibrillation* kept) (Table 1). We mapped diseases to clinical themes, finding that the most strongly associated ones were either T2D complications or known comorbidities (Zghebi et al., 2020; Pearson-Stuttard et al., 2022). We also looked at the most strongly correlated medications, and found that the most strongly correlated ones have indications for severe T2D patients (Medscape, b), heart failure (Medscape, d), erectile dysfunction (Medscape, c) or coronary heart disease (Medscape, a), which are part of the clinical themes (Table 2). The clinical interpretation of these findings can be found in Section 4.3.

| Clinical theme | Disease | r_u1 | r_u2 | L2 norm |
|---|---|---|---|---|
| **Erectile dysfunction** | Erectile dysfunction | 0.588 | -0.284 | 0.653 |
| **Cardiovascular disease** | Atrial fibrillation | -0.045 | 0.22 | 0.225 |
| | Coronary heart disease | -0.0 | 0.211 | 0.211 |
| | Heart failure | 0.032 | 0.187 | 0.19 |
| **Renal failure** | Chronic renal failure | 0.099 | 0.216 | 0.238 |
| | Acute renal failure | 0.003 | 0.206 | 0.206 |
| **T2D complications** | Diabetic retinopathy | 0.329 | 0.219 | 0.395 |
| | T2D with neurological complications | 0.187 | 0.152 | 0.241 |
| | Diabetic polyneuropathy | 0.178 | 0.148 | 0.231 |
| | Diabetic nephropathy | 0.154 | 0.129 | 0.2 |
| **T2D without complications** | T2D without complications | -0.003 | 0.353 | 0.353 |

*Table 1.* Correlation between present comorbidities and u1, u2, in descending order based on L2 norm.

| Indication | Medication | r_u1 | r_u2 | L2 norm |
|---|---|---|---|---|
| **Cardiovascular** | Aspirin | 0.125 | 0.026 | 0.127 |
| | Bisoprolol | 0.062 | 0.111 | 0.127 |
| | Simvastatin | 0.082 | -0.043 | 0.093 |
| | Furosemide | 0.052 | 0.075 | 0.092 |
| | Clopidogrel | 0.056 | 0.069 | 0.088 |
| **Diabetes** | Glucose testing strips | 0.151 | 0.043 | 0.157 |
| | Insulin | 0.121 | 0.096 | 0.154 |
| | Metformin | 0.141 | -0.024 | 0.143 |
| | Diabetes lancets | 0.123 | 0.015 | 0.124 |
| | Gliclazide | 0.099 | -0.003 | 0.1 |
| **Infection** | Amoxicillin | 0.044 | -0.081 | 0.092 |
| **Urological** | Sildenafil | 0.177 | -0.064 | 0.188 |
| | Tadalafil | 0.122 | -0.053 | 0.133 |

*Table 2.* Correlation between present medication prescriptions and u1, u2, in descending order based on L2 norm.

### A.3.3. NUMBER OF PATIENT CLUSTERS

We iterated across different number of clusters $k$ using $(u_1, u_2)$, and 20 different random seeds, and calculated the within-cluster sum of squares (WCSS), which measures the total variation within each cluster. Both k=3 and k=4 were the most stable and robust across seeds, so we used the elbow method (Thorndike, 1953) to choose k=4 (Figure 10).

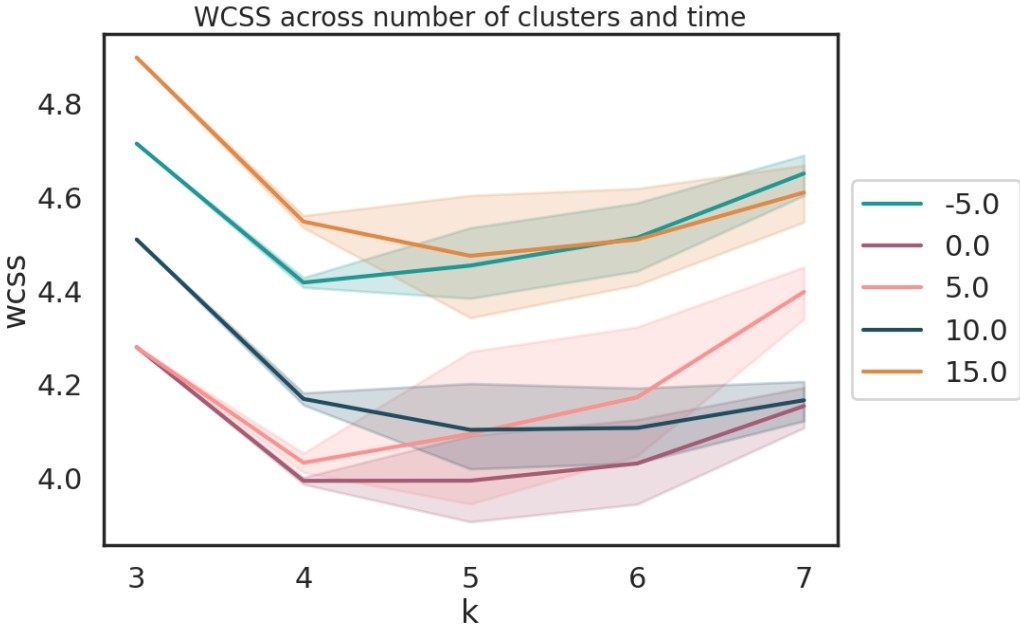

*Figure 10.* Within-cluster sum of squares (WCSS) for each time point, across number of clusters k. The lower the better, as it implies less variation within cluster.

In Table 3 we present the demographics for the final patient groups.

| Cluster | Size | Age at diagnosis (years) | Biological sex (%) | | Ethnicity (%) | | | | | | |
|---|---|---|---|---|---|---|---|---|---|---|---|
| | | | Female | Male | Asian | Black | E.Asian | Mixed background | S.Asian | Unknown | White |
| 0 | 5228 | 59.43 | 34.28 | 65.72 | 0.86 | 1.87 | 0.19 | 0.45 | 5.59 | 0.72 | 90.33 |
| 1 | 1672 | 56.17 | 9.57 | 90.43 | 1.22 | 2.32 | 0.31 | 0.55 | 5.13 | 0.43 | 90.04 |
| 2 | 5579 | 64.42 | 48.43 | 51.57 | 0.75 | 2.24 | 0.15 | 0.4 | 4.44 | 0.38 | 91.65 |
| 3 | 4917 | 56.43 | 50.17 | 49.83 | 0.73 | 2.07 | 0.33 | 0.85 | 5.7 | 0.48 | 89.84 |

*Table 3.* Demographic information for each group: size, age at diagnosis (years), and self reported biological sex and ethnicity (%) when joining the UKBB.

