# OpenReview forum: "Interpreting deep embeddings for disease progression clustering"
_ICML.cc/2023/Workshop/IMLH — IMLH 2023 PosterShortPaper_

### Official Review · Reviewer_hhZf · 2023-06-14
**An interesting research on large population cohort**

**Rating:** 7
**Confidence:** 4

**Review:**

In this manuscript, the authors analyzed and visualized the embedding space of a transformer that is trained on UKBB type-2 diabetes data. Despite the relatively simple technique, the manuscript would still be of interest to the community due to the large scale of the study and the high clinical relevance of the task.

Pros:
1. This study is built on large-scale UKBB dataset.
2. The task is of high clinical relevance.
3. The major methodology is discussed in adequate details.

Cons:
1. There is a lack of detailed description on the de-confounding/preprocessing of the data. Without proper adjustment to the data, the result may be biased by confounders.

---

### Official Review · Reviewer_hJWZ · 2023-06-15
**This paper propose a novel method for interpreting the embedding space of transformer-based models. The results are clinically significant and provide convincing evidence for diagnosis.**

**Rating:** 7
**Confidence:** 3

**Review:**

### Strengths
- The paper is well-written with clear illustrations.
- The experiments are extensive and solid.

### Minor issues:
- lack of reviewing the existing works for interpreting deep embeddings.

---

### Official Review · Reviewer_MZaV · 2023-06-15
**This paper proposes a method for evaluating and interpreting deep embeddings for disease progression using electronic health records (EHRs) from the UK biobank.**

**Rating:** 7
**Confidence:** 5

**Review:**

This paper is well-written and proposed a novel strategy for visualizing and understanding disease progression from electronic health records. However, there are also some limitations in the sense that certain medications from EHRs are correlated with the disease and symptoms, and that well-established procedures exist for prescribing medication as a function of disease.

Pros:

•	The sample size from the UK biobank is impressive

Cons:

•	It would be informative, if the authors could elaborate on the features (i.e. words) that are predictive of a given disease and/or medication, besides the actual words for medication and disease. Are there certain types of descriptions that are informative in terms of predicting the disease progression? Are there any potential biases in terms of site, doctor or ontology?

•	The authors mention in Figure 5 that the data shown is simulated. However, there is no description in the text or appendix of how this was carried out.

•	For a 4-page paper, too much of key information is added to the appendix. Should have considered to expand the paper to a 8-page format instead.

---

### Official Review · Reviewer_cSvW · 2023-06-18
**Using transformer embeddings to analyze disease progression**

**Rating:** 6
**Confidence:** 3

**Review:**

This paper proposes a new method for disease progression clustering, using transformer-based embeddings derived from large-scale structured EHR data, which enables to identify disease progression stages. Evaluations and visualizations demonstrate its effectiveness and look very interesting and intuitive.

---

### Meta-Review · Area_Chair_SpVT · 2023-06-19

**Recommendation:** Accept (Poster)
**Confidence:** 4

**Metareview:**

This paper proposes a method for evaluating and interpreting deep embeddings for disease progression using electronic health records (EHRs) from the UK biobank.

Based on the reviews, all reviewers found the paper interesting, timely, and relevant to the workshop. They unanimously recommended acceptance. Therefore, the AC recommends acceptance of the paper and encourages the authors to address the issues raised in the reviews.

---

### Decision · Program_Chairs · 2023-06-20

Accept (Poster Short Paper)